# COVID Vaccine-Associated Myocarditis in Adolescent Siblings: Does It Run in the Family?

**DOI:** 10.3390/vaccines10040611

**Published:** 2022-04-14

**Authors:** Julia Moosmann, Thomas Gentles, Christopher Occleshaw, Bryan Mitchelson

**Affiliations:** 1Green Lane Paediatric and Congenital Cardiac Service, Starship Children’s Hospital, Auckland 1023, New Zealand; julia.a.moosmann@gmail.com (J.M.); tomg@adhb.govt.nz (T.G.); 2Paediatrics, Child and Youth Health, University of Auckland, Auckland 1023, New Zealand; 3Department of Cardiology, Auckland City Hospital, Auckland 1023, New Zealand; chriso@adhb.govt.nz

**Keywords:** COVID-19, myocarditis, children, vaccination

## Abstract

The development of myocarditis after receiving messenger RNA vaccination against COVID-19 is well documented, particularly in adolescent and young adult males. We report a case of vaccine-associated myocarditis in adolescent brothers following their second dose of the BNT162b2 mRNA vaccine (Pfizer-BioNTech, Mainz, Germany). This report illustrates the need to better understand the mechanisms leading to myocarditis after mRNA vaccination.

## 1. Introduction

There is growing evidence for the development of myocarditis following vaccination with messenger RNA (mRNA) vaccines against COVID-19. This is a rare phenomenon, with a reported incidence of 2.13 cases per 100,000 vaccinations [1,2].

This association is highest within the first week following the second dose, typically affects adolescent and young adult males and commonly presents with symptoms of chest pain, shortness of breath or palpitations [1,3]. The diagnosis of myocarditis is based on the Brighton Collaboration case definition, which includes clinical features combined with electrocardiogram (ECG), cardiac biomarkers, echocardiography and cardiac magnetic resonance imaging (MRI) [4]. The overwhelming majority of cases demonstrate a clinically benign course with complete recovery [5,6]. 

In August 2021, young people aged over 12 years became eligible for the BNT162b2 mRNA vaccine (Pfizer–BioNTech) in New Zealand. This is the only COVID vaccination currently available and approved for use in adolescents in New Zealand. Vaccine uptake within this group has been high, with a vaccination rate of 80% nationally [7]. From our experience, vaccine-associated myocarditis has occurred at similar rates to those reported elsewhere.

The mechanisms causing myocarditis in this context remain poorly understood [8]. A potential mechanism is activation of an innate or acquired immune response directed against the nucleoside-modified mRNA that codes for the spike glycoprotein of SARS-CoV-2, leading to a proinflammatory cascade [9,10]. A genetic predisposition to such a reaction has been postulated, potentially increasing the risk of developing myocarditis post vaccination in particular individuals [11]. Here, we present a case of vaccine-associated myocarditis in two brothers, which may support this hypothesis. 

## 2. Case 1

Patient 1 is a 14-year-old male who presented to his general practitioner with acute-onset left-sided chest pain three days after receiving the second dose of the Pfizer–BioNTech COVID-19 vaccine. The patient was previously healthy with no comorbid conditions and no significant family history of cardiac disease. 

In accordance with local guidelines, patient 1 was assessed for possible myocarditis [12]. High-sensitivity troponin T (hs-Troponin T) was elevated at 81 ng/L (normal reference range <15 ng/L), and patient 1 was referred for specialist assessment. There were no symptoms of COVID infection, and a polymerase chain reaction (PCR) test for COVID-19 was negative. There was no history of previous COVID infection. 

A 12-lead ECG demonstrated normal sinus rhythm with an incomplete right bundle branch block. There were non-specific repolarization abnormalities, including T-wave inversion in V2-V3 (Figure 1A). 

At 8 h, hs-Troponin T had increased to 114 ng/L. An echocardiogram showed a structurally normal heart with normal ventricular dimensions and normal function. Global longitudinal strain was normal. Cardiac MRI 5 days following vaccination confirmed active myocarditis by Lake Louise criteria, including myocardial oedema on T2-weighted imaging and non-ischemic myocardial injury on late gadolinium-enhanced imaging [13]. These changes were focal, involving the inferior basal third of the left ventricle and the posterior obtuse marginal surface (Figure 2A).

Patient 1 was observed in hospital for 3 days. There was no arrhythmia, and the chest pain resolved. Hs-Troponin T peaked at 161 ng/L 5 days following vaccination and decreased to 9 ng/L by day 7 (Figure 3). There were no cardiac symptoms at one- and three-month follow-up. The ECG changes had resolved (Figure 1B), and the echocardiogram was normal. Demographic details, history and laboratory findings are illustrated in Table 1. 

## 3. Case 2

Patient 2 is a 12-year-old male with no previous medical history or comorbid conditions and is the younger sibling of patient 1 (details in Table 1). Patient 2 presented one week following his brother’s admission with acute left-sided chest pain that developed 48 h after receiving the second dose of the Pfizer–BioNTech COVID-19 vaccine. Patient 2 displayed no symptoms of COVID infection, and a PCR test for COVID-19 was negative. There was no history of previous COVID infection. Hs-Troponin T on admission was elevated at 141 ng/L. The 12-lead ECG was normal. 

An echocardiogram on admission demonstrated mildly impaired global longitudinal strain of −17.6% (lower limit of normal for age, −19.2%) [14] with a normal ejection fraction (67%). Cardiac MRI 5 days following vaccination was similar to that of the older sibling. There was a large region of full-thickness myocardial oedema in the inferior and posterior obtuse marginal surfaces of the heart. Late gadolinium-enhanced imaging confirmed the presence of non-ischemic myocardial injury in these regions (Figure 2B).

Patient 2 was observed for a total of 5 days, during which hs-Troponin T peaked at 451 ng/L 4 days following vaccination and fell to 5 ng/L by day 8 (Figure 3). Chest pain persisted for several days but had resolved 48 h prior to hospital discharge. There was no arrhythmia.

There was no recurrence of chest pain at one- and three-month follow-up. ECG and echocardiogram were normal, with normalization of global longitudinal strain.

## 4. Discussion

In this report, we described myocarditis occurring after the mRNA Pfizer vaccination in two adolescent siblings, highlighting the need to better understand the mechanisms behind vaccine-associated myocarditis and potential predisposing factors. 

There is little in the current literature to suggest an increased risk of vaccine-associated myocarditis in first-degree relatives. Consequently, there is no recommendation as to whether it is safe to proceed with vaccination if a first-degree relative has already been affected [12]. For this reason, vaccination was recommended for the younger brother. 

The clinical symptoms of vaccine-associated myocarditis described include chest pain, shortness of breath, tachycardia and palpitations, with a median symptom onset of two days following vaccination [5,15]. Both siblings developed symptoms within four days after receiving their second dose of Pfizer-BioNTech COVID-19 vaccine and presented with acute left-sided chest pain. Both received symptomatic treatment with non-steroidal anti-inflammatory medication and clinically improved, with resolution of chest pain and normalization of serum troponin level. ECG and echocardiogram changes had resolved at one-month and three-month follow-up. 

Elevated troponin is a key finding in cases of vaccine-associated myocarditis; however, it is not known whether this correlates with the degree of inflammation or with prognosis. In one study of childhood myocarditis, troponin levels were not strongly predictive of outcome [16]. In contrast, in adult patients with myocarditis secondary to COVID-19 infection, troponin levels were higher in those with more severe COVID-19 infection [17]. Whether this can be extrapolated to vaccine-associated myocarditis is unknown. We noted higher hs-troponin T levels in the younger brother, but there was no significant difference in the severity of his clinical condition or in the speed of recovery, although the MRI changes were more extensive compared to those of the older brother.

Echocardiographic findings in vaccine-associated myocarditis include left ventricular dysfunction, abnormal longitudinal or circumferential myocardial strain and pericardial effusion [18]. Our first patient had a normal echocardiogram, but the second patient had mild impairment in global longitudinal strain. Cardiac MRI is more sensitive than echocardiography in detecting myocarditis [19]. For this reason, all young patients presenting with symptoms and elevated troponin after COVID vaccination are referred for cardiac MRI, whether or not the echocardiogram is abnormal. Both siblings in this report had definite changes in cardiac MRI consistent with active myocarditis according to the Lake Louise criteria [13] and of a type that has been previously described in vaccine-associated myocarditis [19,20,21]. 

This presentation of vaccine-associated myocarditis illustrates the need to better understand the mechanisms of this entity. Our case supports a potential genetic susceptibility; however, statistically this could also be explained by random chance. We considered alternative causes of myocarditis in our cases, including autoimmune conditions and typical viral causes. Investigations for these were negative in both siblings. Genetic background in response to environmental factors may play a role in the immune response seen in patients with myocarditis [22,23,24], such as a dysregulated cytokine response (e.g., IL-1beta, IL-17 and TNF-alpha), leading to infiltration of the myocardium [25]. Other proposed genetic mechanisms include alterations in structural proteins, creating a more vulnerable myocardium [22]. Potential mechanisms for vaccine-associated myocarditis include cytokine-induced hyperinflammatory response, development of autoantibodies and molecular mimicry of the spike glycoprotein of SARS-CoV-2 [9,10]. A recent report by Won et al. demonstrated higher interleukin levels (IL-18) in a myocardial biopsy of a patient with myocarditis following COVID-19 mRNA vaccination [9]. Cross-reactivity of antibodies against alpha myosin and transglutaminase has been identified in COVID infection. Vojdani et al. indicated that a similar response against the spike protein of the vaccine may lead to vaccine-induced autoimmunity [26]. However, these reports are limited, and further research into this is required.

A better understanding of the mechanisms of vaccination-associated myocarditis—particularly the potential for genetic predisposition or the autoimmune response—may help guide future recommendations. 

## 5. Conclusions

This is the first reported presentation of vaccine-associated myocarditis in siblings. This report supports a possible genetic susceptibility and illustrates the need to understand the mechanisms behind this rare cause of myocarditis. 

## Figures and Tables

**Figure 1 vaccines-10-00611-f001:**
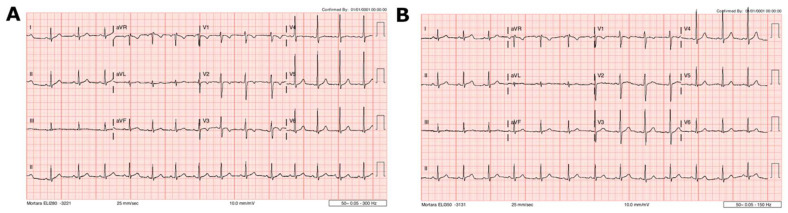
Case 1 ECG findings. (**A**) ECG at presentation demonstrating abnormal repolarization, including T-wave inversion in V2–V3 and T-wave flattening in V4. (**B**) One month after presentation, showing resolution of these changes.

**Figure 2 vaccines-10-00611-f002:**
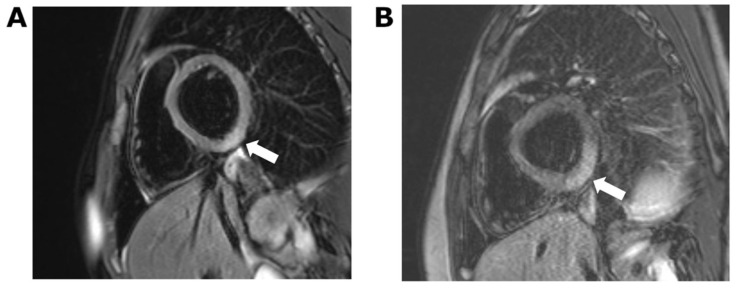
Cardiac MRI showing short-axis T2-weighted image with fat saturation. (**A**) Case 1: ill-defined area of increased signal in the inferior left ventricular myocardium indicative of focal oedema. (**B**) Case 2: more extensive region of oedema in comparison to case 1, which extends into the posterior obtuse marginal surface.

**Figure 3 vaccines-10-00611-f003:**
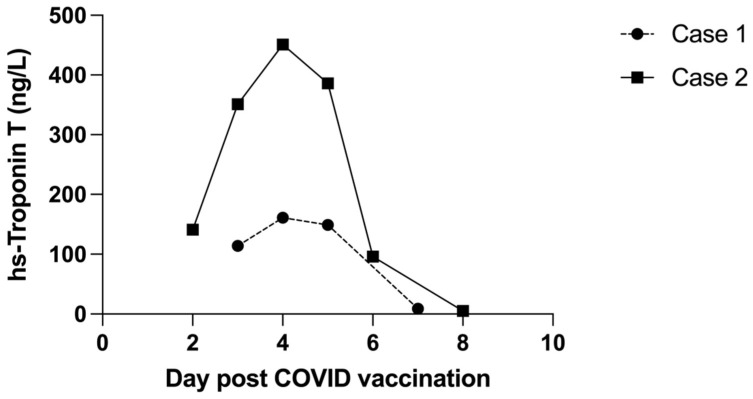
Comparison of trends in hs-Troponin T during the acute phase of myocarditis.

**Table 1 vaccines-10-00611-t001:** Demographics and laboratory findings.

	Patient 1	Patient 2
**Patient anthropometrics**		
Age (years)	14	12
Weight (kg)	55.3	54.5
Height (cm)	164	162
BMI (kg/m^2^)	20.4	20.8
**Vital parameters at presentation**		
Heart rate (bpm)	88	85
Blood pressure (mmHg)	117/79	103/58
SpO_2_ (%)	98	99
**Laboratory parameters at presentation**		
White blood cell count (E + 9/L)	5.39	5.57
Lymphocyte count (E + 9/L)	2.68	2.23
C-reactive protein (mg/L)	4.6	1.5
Haemoglobin (g/L)	143	138
**COVID-19 history**		
	No previous recorded COVID-19 infection Parents received COVID-19 Pfizer-BioNTech vaccination without side effects	

## Data Availability

Data is contained within the article.

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
