# Peer review of "COVID Vaccine-Associated Myocarditis in Adolescent Siblings: Does It Run in the Family?"

_vaccines, 2022, doi:10.3390/vaccines10040611_

Round 1
Reviewer 1 Report
The topic of the case report „COVID vaccine-associated myocarditis in adolescent siblings – does it run in the family?“, represents a modest but relevant contribution to the emerging field of genetic makeup involvement in COVID and COVID-related issues, such as the vaccination, and its potential side effects. In their well written paper authors have described two adolescent siblings that developed myocarditis following the administration of the second dose of Pfeizer-BioNTech vaccine, stressing the possibility that genetics played a role in the occurrence of the condition.
I have only some minor criticisms/comments
The two patients should be better described. For the older brother there is only a statement “previously healthy”, while for the younger even that sentence is lacking. There should be more details. Although at the moment of clinical examinations the two patients had negative COVID PCR tests, what about their previous COVID status. It should be made clear that they never had the infection (or if they had it, it was mild, severe...?). What about obesity/BMI, family history of cardiac conditions, etc.- additional information are needed.
Since the submission of the present manuscript, several papers dealing with the link between vaccine and myocarditis have been published and should be added to the reference list. For instance:
Chin et al doi: 10.1007/s00246-022-02851-x
Won et al doi: 10.3389/fimmu.2022.851620.
Author Response
We have included more details about both patients in our revised manuscript, including background information in the new Table 1. We have included both suggested references and discussed the findings of Won et al. in our discussion section.

Reviewer 2 Report
The case report about COVID vaccine-associated myocarditis in adolescent siblings is impartant to understand the side effect of COVID vaccine. The author mentioned the two brothers were vaccine-associated myocarditis, and I think the topic is really interesting. I suggest more genetic factors associated myocarditis should be added in the discussion.
Author Response
We have included more information about genetic factors associated with vaccination induced myocarditis and myocarditis in general in our discussion.
Reviewer 3 Report
- The risk of myocarditis in children or adolescents immunized with COVID-19 vaccine is 2.13 out of 100,000, which can occur in two brothers of the same family, indicating a strong genetic link.The clear description of this situation provides guidance for the use of COVID-19 vaccine in the future, and also provides a preliminary direction for the study of the association between COVID-19 vaccine and myocarditis.
- Please expand and detail the background of myocarditis associated with COVID-19 vaccine immunization.
Author Response
Thank you for your comments. We have included more background information of myocarditis associated with COVID vaccination in our introduction.
Reviewer 4 Report
Thank you for giving me an opportunity to review this manuscript. Myocarditis after RNA vaccination against COVID-19 have been well documented, particularly in adolescent and young adult males. The authors reported a case of vaccine-associated myocarditis in two brothers, proposing a possible mechanism behind vaccine-associated myocarditisIn. However, this needs more studies to demonstrate the reliability of this hypothesis. In total, this is an interesting case report brings new sights to explore the causes of this rare phenomenon. I suggest accepting it directly.
Author Response
Thank you for your positive review of our manuscript.
Reviewer 5 Report
This is an interesting case report, well, sib-case report. It would be beneficial to know if parents had any symptoms (I assume not, but it would have been great had you managed to assess them too). Any history of cardiac disease in the family? Risk assessment of parents, commensurate for their age? Grandparents? Please try to provide more context for these claims (although, I fear you may not find some ground-breaking elements, but it would be good to report those). The paper indeed supports the possible genetic link, but then again, this is a rare occurrence, and there were many more that did not have sib-related risk, so one might also claim this to be a random event that you managed to pick up (I see no way to exclude the hypothesis that this was a truly random coincidence, especially given the sheer number of vaccinated children, where the sib-pairs did not have co-occurrence). Luckily, the extent of myocarditis was mild (at least for now; you should perform regular or even more-frequent than regular check-ups in the future). The manuscript is worth publishing, having this listed as a potential side-effect. Also, do mention the need to develop an international database of such events, in case that someone tries to pool those and create a predictive monogenic database for side-effects risks.
Author Response
We have included more details about both patients and family history in the revised manuscript including background information in the new Table 1. We have also included a sentence in our discussion that there might be a statistical chance of two siblings getting myocarditis by random chance.
Since the manuscript has been submitted, both patients had further follow-up appointments demonstrating good clinical health and no echocardiographic signs of ventricular impairment (strain, EF etc.). This has been included in the case presentations.
Reviewer 6 Report
Overall, this is a relatively well-written manuscript. Some minor edits:
1-Within the second paragraph of the Case 1 section, there is a sentence that reads: "In accordance with local guidelines, he was assessed for possible myocarditis." Change, "he," to, "Patient 1."
2- Within the second paragraph of the Case 1 section, there is a sentence that reads: "A high-sensitivity Troponin T (hs-Troponin T) in the community was elevated at 81 ng/L (normal reference range <15 ng/L) and he was referred promptly for specialist assessment." This sentence is cumbersome......it's not clear why the phrase, "in the community," is included. Further, rather than using the pronoun, "he," be more specific and use, "Patient 1."
3-In the fourth paragraph of the Case 1 section, the authors refer to the Lake Louise criteria. Please include a reference for these criteria.
4-Begin the fifth paragraph of the Case 1 section with, "Patient 1," rather than, "The patient."
5-Ideally, please move the figures to the section after the Case 2 section and before the Discussion section. As laid out currently, it's somewhat awkward and confusing as the figures in some cases (i.e. Figures 2 and 3) refer to both Case 1 and Case 2. (Note that it is recognized that this may not be something the authors have control over.)
6-Within the first paragraph of the Case 2 section, there are two sentences that read: "He presented one week following his brother's admission with acute left sided chest pain that developed 48 hours after receiving the second dose o of Pfizer-BioNTech COVID-19 vaccine. He displayed no symptoms of COVID infection and a PCR test for COVID-19 was negative." Rather than using the pronoun, "he," be specific and use the term, "Patient 2." Hyphenate, "left sided," so that it is, "left-sided."
7-Begin the third paragraph of the Case 2 section, with "Patient 2," rather than "The patient."
8-Change the first sentence of the last paragraph of the Case 2 section so that it reads: "One month following presentation there was no recurrence of chest pain."
Author Response
We have implemented all the recommended changes in our revised manuscript.